# An Assessment of Water Supply Governance in Armed Conflict Areas of Rakhine State, Myanmar

Thin Khaing and Thi Phuoc Lai Nguyen *

Department of Development and Sustainability, School of Environment, Resources and Development,
Asian Institute of Technology, Klong Luang, Pathum Thani 12120, Thailand
* Correspondence: phuoclai@ait.asia

**Abstract:** This study aims to analyze the actors and institutions for public water supply governance in armed conflict areas of Rakhine State, Myanmar. Using Stakeholder Salience Theory and Institutional Analysis of data collected from four participatory workshops and interviews with 160 water stakeholders from the four townships in Rakhine State, the findings revealed that although the water supply system is managed and governed by the state water authorities with the involvement of many administrative, political, and sectoral technical agencies and organizations, the non-formal community organizations such as ethnic armed military and religious institutions also have a strong interest in water supply and are considered dangerous actors in the water supply governance process. Diverse water actors held different perspectives and perceptions of water supply quality and quantity because of their different power holdings and political and economic interests. The state actors seemed biased on their positive performance, demonstrating their satisfaction with the current water supply governance, while community, private sectors, and household water users instead showed their dissatisfaction with the quality and quantity of the current water supply system, but they stayed neutral about the water supply governance performance. The research showed the complexity and dynamics of water actors' powers and interests in armed conflict areas. In addition, there is a lack of socio-technical and financial capacity for the investment and maintenance of water distribution and collection infrastructure and facility, as well as water quality and quantity monitoring and evaluation. The study appeals to the development and peacebuilding organizations working in conflict areas to promote adaptive governance for community learning and adaptation to social-political and environmental change over time.

**Keywords:** public water supply governance; stakeholder salience theory; actors and institutions analysis; armed conflict; Rakhine State; Myanmar



## 1. Introduction

Water governance is a broader concept than water management [1], and it refers to "the range of political, social, economic and administrative systems that are in place to regulate development and management of water resources and provisions of water services at different levels of society [2]". As stakeholder participation and empowerment is one of the key elements of good water governance [3], it is a necessity to create an enabling environment for stakeholder involvement and the combined commitment of government and various groups in civil society, particularly at local/community levels, as well as the private sector, to achieve the effective water governance for addressing problems of water supply [4]. The role of self-regulated governance plays an important role in a difficult context, whose policy design is tailored to facilitate user autonomy and strengthens user self-regulated governance [5]. Adaptive governance systems often self-organize as social networks with teams and actor groups [6]. Sound water resource management systems lead to the sustainability of water resources and well being of the people in the country [7]. The

relationship between water management and conflict has complexity, which is not because of access to water but related to the way people manage the system of water supply [8].

Previous research findings showed "there are weak water management systems in conflicting areas globally [9]", as "armed conflicts directly or indirectly affect water governance and management systems [10]". Due to the lack of coordination and participation of multi-stakeholders and the public, the implementation of water policy is a top-down water management structure [11], and "water institutional structure formation is not very common in water management [12]".

Although Myanmar is rich in water resources, it possesses 12% of the whole of Asia's freshwater resources and 16% of the ASEAN nations. The growing pressure on the existing water resources is the uneven temporal and spatial distribution of water resources, creating further challenges for water allocation [13]. In addition to that, the current policy and administration of water resources in Myanmar are scattered and unfocused, and overlapping interests lead to unclear jurisdiction [14].

Many of the issues related to water governance have never been addressed adequately in Myanmar due to the mismanagement of water resources for a long time. Water resources in Myanmar are in a favorable situation, as its water per capita is more than all surrounding countries; however, the availability of freshwater supply depends on reservoirs, communal ponds, and private collection of rainwater and groundwater. The current status of the water supply system in Myanmar still lacks the required infrastructures, supply network, and water resources depletion due to drought and climate change, and also many agencies are engaging in water supply and management without proper cooperating and coordinating with each other, while there are also long-standing conflicts between the Myanmar Armed Forces (the Tatmadaw) and various insurgent groups known as Ethnic Armed Organizations (EAOs) [14].

Rakhine State in Myanmar is located in the tropical area with abundant and concentrated rainfall during the rainy season, while the dry seasons last long with a considerable evaporation rate, resulting in a disproportional temporal distribution of water quantity in natural ponds, which are the main sources of drinking water supply while saltwater intrusion into surface water from many river networks within basin [11,15].

According to UNDP's Local Governance Mapping [16], providing safe and equitable access to drinking water is a core responsibility of the government at the local level. However, government authorities in Rakhine State have only recently begun to invest more resources in this sector, while at the same time, residents of Rakhine State's urban and rural communities have an urgent need for safe drinking water.

As stated by the International Crisis Group [17], Rakhine State is a site of active conflict with frequent clashes between the Arakan Army and the Tatmadaw near civilian areas due to the emergence of the Arakan Army insurgency in Rakhine State from around 2015, and its dramatic escalation since early 2019 was neither inevitable nor unforeseeable. Since January 2019, the conflict has intensified between the Arakan Army and the Tatmadaw, and it is continuing. As a result of this armed conflict, the northern and central townships of Kyauktaw, Mrauk U, Rathedaung, and Buthidaung have been most affected by both armed conflict and recent communal conflict in Rakhine State so that many IDPs (Internally Displaced Persons) from rural areas were hosted in the town wards of these affected townships [17].

Although there might be previous research on water supply governance, and many documents have been published stating good water governance is a prerequisite to improving water management [3,18] in the world, there are still some knowledge gaps to know on (1) Who are the key actors and institutions engaging and influencing in the governance network of the public water supply service in the most armed conflict-affected areas of Rakhine State? (2) What are the different types of those key actors and institutions along with their level of saliency during the conflict situation? (3) What are the perceptions of key stakeholders related to the governance of the public water supply and their relationships during the armed conflicting period? (4) How do these key stakeholders of public water

supply governance organize the institutional arrangements to be effective public water supply management in Rakhine State? and lastly (5) What is the role of the community's self-regulated water supply system in Rakhine State?

In light of such knowledge gaps, the main research problem was raised whether currently engaging actors and institutions are positively contributing to public water supply governance in the context of armed conflict.

Towards solving the main research problem, the decision model was set as if and when the institutional arrangements of water supply key actors and institutions are positive, it is likely to achieve a high level of adaptive and good public water supply governance.

Thus, the overall objective of this research was to analyze the roles and engagement of state and non-state key actors and formal and informal institutions for public water supply governance in armed conflicting areas of Rakhine State, Myanmar.

The specific objectives of this study were (1) to provide an overview of the public water supply distribution system and water supply governance structure in Rakhine State, Myanmar, (2) to identify the key actors and institutions and classify their types and level of saliency in the governance network of the public water supply service in the most armed conflict-affected areas of Rakhine State, (3) to assess the stakeholders' perception on water supply situation, issues, causes and consequences of policy changes during the armed conflicting period in Rakhine State, and (4) to analyze the multi-stakeholder perspectives for public water supply governance in armed conflicting areas of Rakhine State. The ultimate goal of this study is to recommend a suitable governance model for freshwater supply in armed conflicting areas of Rakhine State.

This study is very important, as water is one of the most essential and indispensable natural resources in armed conflicts to ensure basic access of the population to water, sanitation, and hygiene in the conflicting area. Armed conflicts have devastating impacts on human life and environment [19], along with the resource management problems resulting from governance failures [6,20]. Although there are many reasons why water management fails, the crisis of water governance due to the fragmented institutional structures and weak regulatory framework exacerbate many problems of water availability [21] in the world.

Many problems in water management are more associated with governance failures than with the resource base [2,22] and require significant reforms in water governance by taking into account contextual factors. However, theories and methods for sustainable water resource management and governance are still in the developmental phase, and continuous experiments in the application are required to determine effective approaches for research and practice [23,24].

## 2. Materials and Methods

### 2.1. Study Area

Rakhine State in Myanmar is geographically located in the coastal area that faces the issue of saltwater intrusion into surface water from many river networks within basins [15]. In terms of socio-economic development, Rakhine State is the 2nd poorest state in the country [16], while there are ongoing major armed conflicts from 2018 to 2020. This study focused on the most armed conflicts affecting townships of Kyauktaw, Mrauk U, Rathedaung, and Buthidaung, among the total of 17 Townships in Rakhine State. The study area map is presented in Figure 1.

In Rakhine State, the freshwater resources are mainly from rainwater as surface water storage in natural ponds and man-made small dams for providing the town water supply services by the Department of Municipal/Development Affairs with the support of elected Town Municipal Affairs Committees in each township. However, as stated by the Rakhine State Department of Municipal/Development Affairs, depending on the rainwater storage capacity and the number of population and households, the supply and demand of public water supply services can be different.

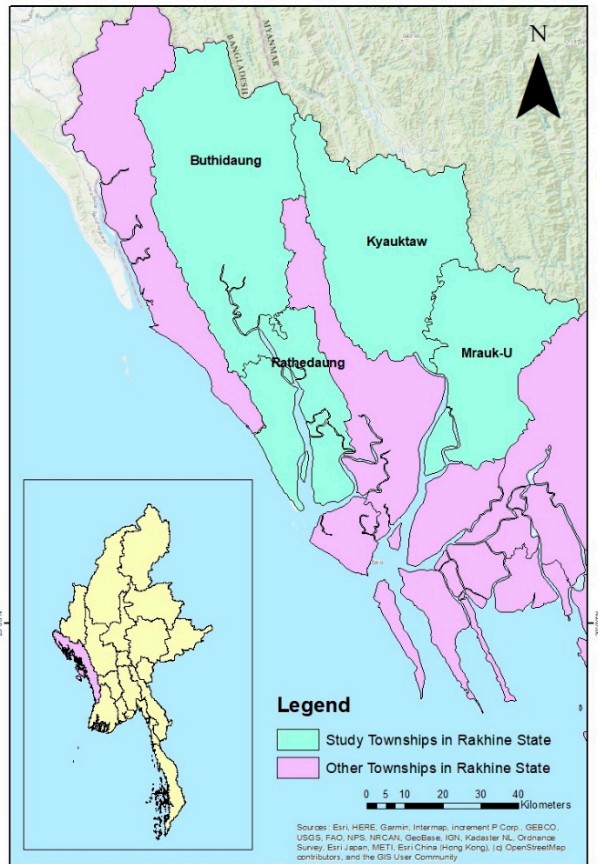

**Figure 1.** Map of Study Area.

The current public water supply system in the study area is a gravity flow system combined with a pumping system from natural water resources of ponds and small dams to the water collection tanks to cover the town wards. Then, using a pipe water distribution system from the water collection tanks to individual tap stands located in user households of urban area town wards.

As the provided town water supply service is scheduled based and rotation basis to cover a minimum of 50% to a maximum of 85% of households, there are also informal private hand-dug wells and deep tube wells established by water users' own arrangement in order to complement the insufficient public water supply with the groundwater from their respective home yard sources.

### 2.1.1. Urban Public Water Supply Distribution System

The common public water supply distribution system found in the study area of Rakhine State is mainly based on a gravity flow water supply system combined with a pumping system from natural water resources of ponds and small dams to the water collection tanks first. Then, different sizes of transportation pipelines are installed to cover the town wards for transporting water from each collection tank. The individual water user households have distributed the fresh water from the water collection tanks using different distribution pipelines. The schematic diagram for the town water supply distribution system in the study area is presented in Figure 2A–C, respectively.

However, the appropriate water purification systems for water quality control were not installed in this current town water supply system in this study area, and there is still lacking water meters installation for the systematic collection of water terrific and efficient and effective utilization of distributed water quantity control.

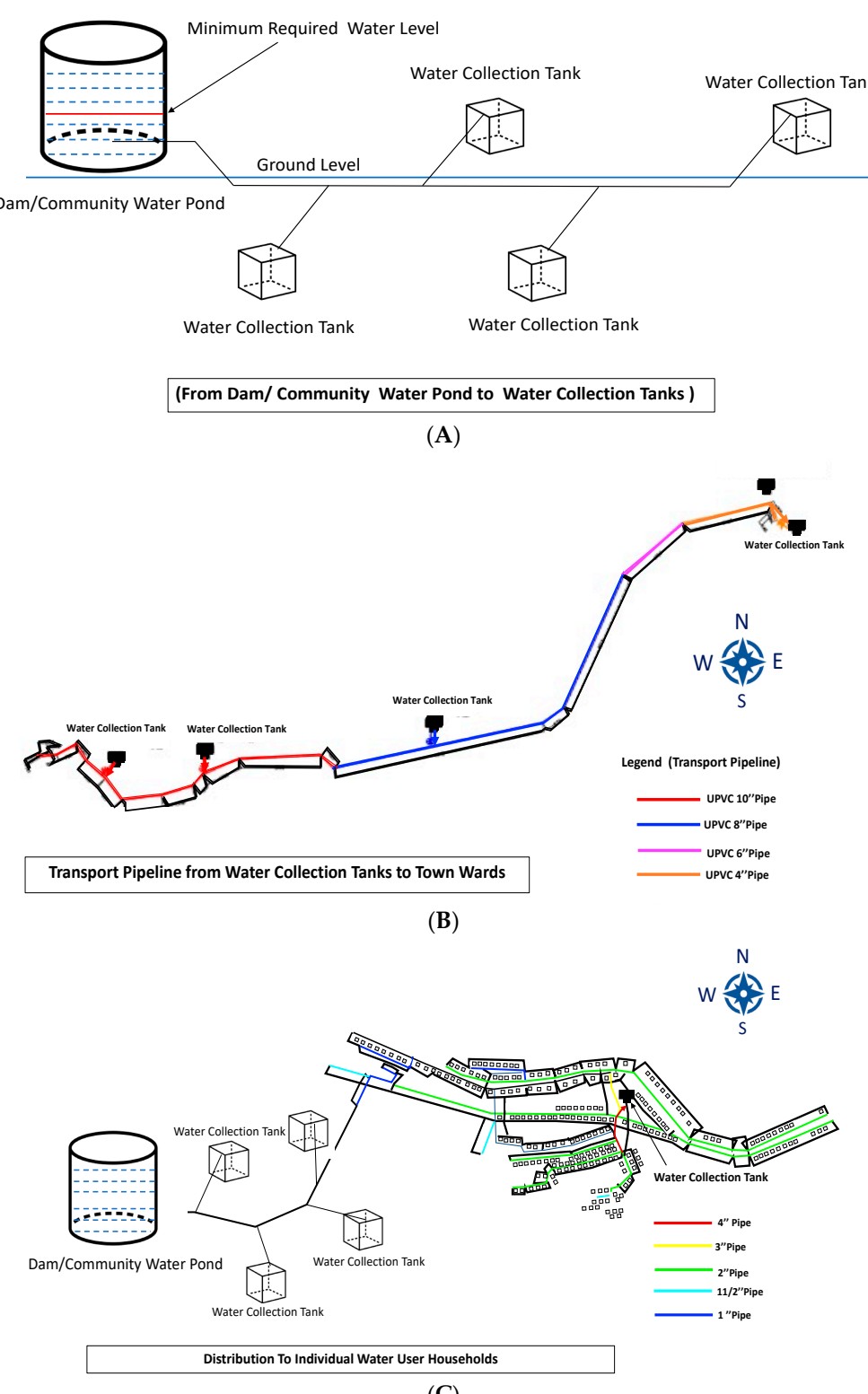

**Figure 2.** (**A**). Schematic diagram of public water supply system (**B**). Schematic diagram of public water supply system. (**C**). Schematic diagram of public water supply system in the study area. Source: Rakhine State Department of Municipal/Development Affairs, 2021.

2.1.2. Urban Public Water Supply Governance Structure

In Myanmar, there was no single agency at the union/central level for urban water supply as fourteen local governments (States/Regions Governments) take responsibility for

providing city and own water supply in their respective administrative areas except the city and towns of Yangon City Development Committee (YCDC), Mandalay City Development Committee (MCDC), and NayPyiTaw Development Committee (NDC).

At the local government level, Rakhine State's township-level public water supply governance structure is presented in Figure 3. In Rakhine State, a total of seventeen townships' public water supply systems have been managed and responsible by Rakhine State Government at the overall State level, while the respective Department of Municipal/Development Affairs and elected Township Municipal Affairs Committees are taking specific responsibilities for urban/town public water supply services. It was important to note that the Township Department of Rural Development is responsible for rural water supply services in each township.

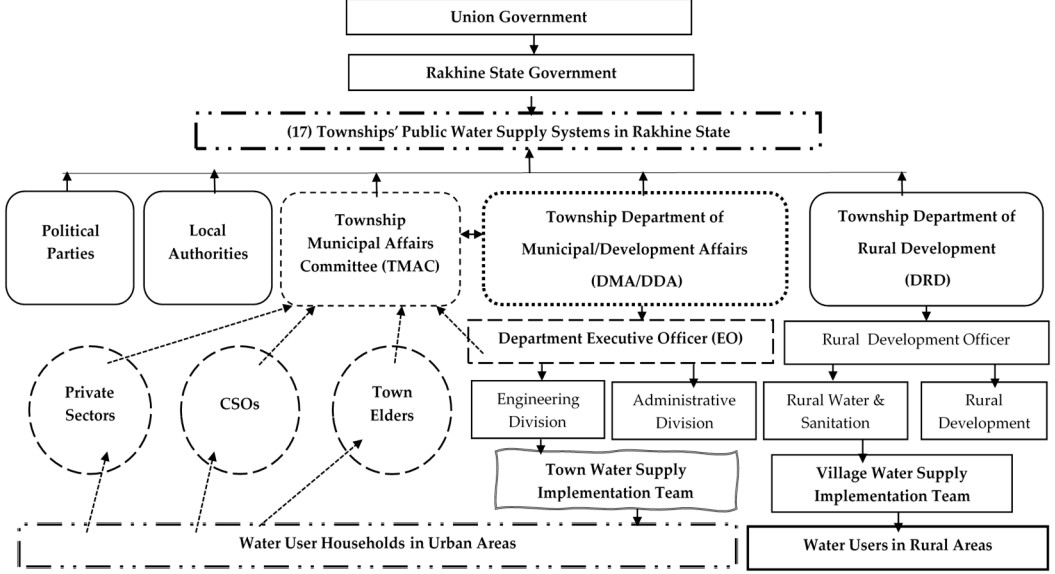

**Figure 3.** Township-level public water supply governance structure. Source: Rakhine State Department of Municipal/Development Affairs, 2021.

In the detailed governance structure of the urban/town public water supply system in the study area, from 2018 to 2020, the Township Municipal Affairs Committees (TMAC) were formed to supervise and manage urban/town water supply service by conducting elections in each township towards the emergence of governance body with five elected committee members.

The secretary of TMAC was not elected, but the respective Township Municipal/ Development Affairs' Executive Officer was appointed as the secretary of TMAC by default. The rest of the committee's members were elected from Town Elder groups, Civil Society Organizations (CSOs), Private Sectors, Businesses, Households, and Water Users, respectively.

In addition to that, the local authorities and political parties, including members of parliament from the respective township, were also part of the public water supply governance structure. The focal government department for town/urban water supply service implementation was the Township Department of Municipal/ Development Affairs, under the direct supervision of the State Department of Municipal/Development Affairs.

### 2.1.3. Theoretical Background

Management and Transition Framework (MTF) [25] is applied to design this research. MTF is a conceptual framework for comprehensive analyses of water management and mainly focuses on the human dimension of water governance and the learning process [26]. It covers three broad thematic areas of (1) adaptive management, (2) social learning, and (3) the Institutional Analysis and Development Framework [27]; thus, the Institutional Analysis and Development Framework (IAD) [28] was integrated into the research design IAD

introduces the concept in which local actors interact to create the institutional arrangements that shape their collective decisions and individual actions [29]. It was used to conceptualize the operational outcomes of institutions affected by the armed conflicts as the result of how water governance actors organize the institutional arrangements in freshwater supply management. This institutional analysis aimed to examine if adaptative freshwater supply governance emerged from the armed conflicting situation and if the actors at the local level organized themselves for adaptive institutional arrangements. Furthermore, Stakeholder Salience Theory [30] was thus applied to map, identify and classify the key actors and stakeholder analysis. The analysis of stakeholders focuses on three attributes of Power, Legitimacy, and Urgency of the Stakeholder Salience Theory to identify and classify the different stakeholder groups for water supply governance.

*2.2. Research Methods*

2.2.1. Data Collection

This study focused on the most armed conflicts that affected townships in Rakhine State from 2018 to 2020. Therefore, out of seventeen townships in Rakhine State, the most armed conflict-affected four townships of Kyauktaw, Mrauk U, Rathedaung, and Buthidaung were selected for primary data collection.

The public water supply distribution system and its governance structure of the study area were collected from Rakhine State governmental departments. Primary data collection was conducted through participatory workshops with multi-stakeholders in four selected townships. Forty participants from each township were invited to the workshop at each township, making the total number of 160 participants from four townships engaged in the participatory workshops. The first part of the workshops was to discuss and identify each stakeholder's roles, responsibilities, and kinds of their involvement in water supply governance. For institutional analysis, interview questions were structured to investigate the different actors' perceptions during the second part of the workshops to examine the stakeholders' perceptions of the current water supply governance in Rakhine, causes and consequences of the inadequate water supply system, inter–intra interaction among water institutions, satisfaction level of access to town water supply services.

2.2.2. Data Analysis

The primary data related to the current situation of freshwater resources and the causes and consequences of institutional policies during the armed conflicting period (2018–2020) were analyzed using narrative policy analysis [31].

A stakeholder analysis was performed following the stakeholder salience theory to identify and classify stakeholders' saliency. Descriptive statistics were used to depict the respondents' socio-economic and demographic profiles. Thematic analysis was applied to analyze the structured interviews on stakeholders' perceptions of the current state of freshwater management in Rakhine. Furthermore, the research indicators and variables used in the structured interviews and questionnaire to collect the stakeholders' perceptions of the water supply governance performance are presented in Appendix A. Based on the IAD framework for actor and institution analysis, these indicators were developed to assess stakeholders' perceptions of water supply governance in the study area.

**3. Results**

*3.1. Socio-Demographic Characteristics of Multi-Stakeholders in the Participatory Workshops*

As presented in Table 1, the Chi-square test results on workshop respondents' gender, age, and occupation were not significant at $p < 0.05$, which means there are no differences among the four townships in terms of the respondents' proportion in gender, age, and occupation categories.

**Table 1.** Multi-stakeholder workshop participants' characteristics in the study area.

|  |  | Kyauktaw | Mrauk U | Rathedaung | Buthidaung | Statistic | df | *p*-Value |
|---|---|---|---|---|---|---|---|---|
| Number of Respondents (*n*) | 160 | 40 | 40 | 40 | 40 |  |  |  |
| Gender (%) | Male | 80.0% | 92.5% | 72.5% | 75.0% | $X^2 = 5.9375$ | 3 | 0.115 |
|  | Female | 20.0% | 7.5% | 27.5% | 25.0% |  |  |  |
| Age (%) | 20–30 | 5.0% | 20.0% | 20.0% | 22.5% | $X^2 = 14.914$ | 9 | 0.093 |
|  | 31–40 | 17.5% | 10.0% | 27.5% | 25.0% |  |  |  |
|  | 41–50 | 15.0% | 25.0% | 20.0% | 20.0% |  |  |  |
|  | >50 | 62.5% | 45.0% | 32.5% | 32.5% |  |  |  |
| Education (%) | Undergrad | 55.0% | 40.0% | 62.5% | 70.0% | $X^2 = 16.971$ | 6 | 0.009 * |
|  | Secondary | 27.5% | 35.0% | 35.0% | 27.5% |  |  |  |
|  | Primary | 17.5% | 25.0% | 2.5% | 2.5% |  |  |  |
| Occupation (%) | Business | 37.5% | 35.0% | 35.0% | 32.5% | $X^2 = 1.3932$ | 6 | 0.966 |
|  | Employee | 27.5% | 27.5% | 32.5% | 37.5% |  |  |  |
|  | Others | 35.0% | 37.5% | 32.5% | 30.0% |  |  |  |

* The result is significant at *p* < 0.05.

However, it was found that Mrauk U township has stakeholders having lower education levels while other townships have 50–70% of participants holding an undergraduate degree. Furthermore, most participants of the workshops were male and had an age over 50 years old. It reveals that the participants from each key stakeholder group represented the majority of older males with university education; however, it seems atypical for the Rakhine population, but male and senior domination is typical in political and administrative positions in Myanmar.

### 3.2. Town Water Supply Stakeholders and Power Dynamics
Public Water Supply Stakeholders and Their Salience

During the multi-stakeholder workshop, eight key organizations and institutions participated in identifying and classifying the types and salience levels of water stakeholders. A total of seven kinds of key stakeholder categories were classified. To determine the individual stakeholder' attributes of *power*, *legitimacy*, *and urgency* with the consensus of all participants during the workshop, all the workshop participants were asked: which of the following statements appear to describe best how your Township's public water supply service deals with the claims of different stakeholder groups?

1. In your Township Public Water Supply Service, which stakeholder groups having urgent claims get the highest priority and attention?
2. In your Township Public Water Supply Service, which stakeholder groups having legitimate claims get the highest priority?
3. In your Township Public Water Supply Service, which stakeholder groups having urgent and legitimate claims get the highest priority?

Based on the results of the above questions, the different types of stakeholder groups were identified and classified, as presented in Table 2 and Figure 4.

**Table 2.** Classification of public water supply stakeholders in Rakhine State by using Stakeholder Salience Theory.

| Type of Stakeholders | Possessed Attributes | Saliency | Stakeholder Groups |
|---|---|---|---|
| Dormant | Power | *Low* | Religious Organizations * |
| Discretionary | Legitimacy | *Low* | Department of Rural Development, Private Sector |
| Demanding | Urgency | *Low* | Water Users |
| Dominant | Power, Legitimacy | *Moderate* | Military Institutions ** |
| Dangerous | Power, Urgency | *Moderate* | Ethnic Armed Organizations *** |
| Dependent | Legitimacy, Urgency | *Moderate* | Civil Society Organizations |
| Definitive | Power, Legitimacy, Urgency | *High* | TMAC/Town Elders, Department of Municipal/Development Affairs, Local Authorities, Political Parties |

*, **, *** the "Dormant", "Dominant", and "Dangerous" types, in the local context, were identified as the other stakeholder groups of Religious Organizations *, Military Institutions ** and Ethnic Armed Organizations ***.

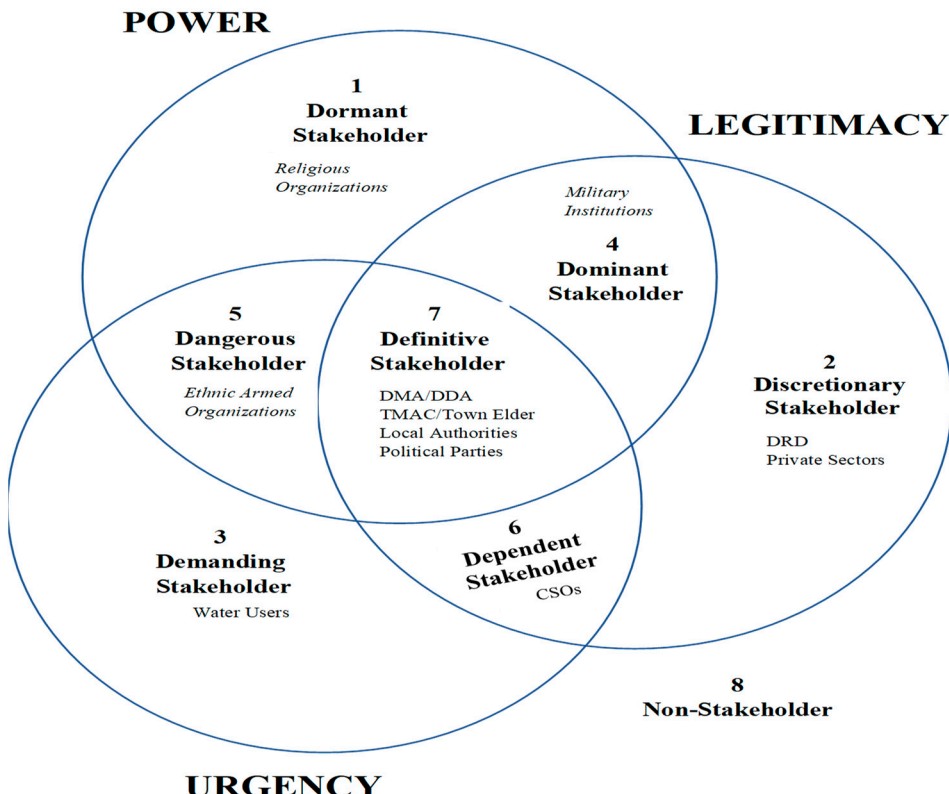

**Figure 4.** Power dynamics of water supply stakeholders in Rakhine State. TMAC = Town Municipal Affairs Committee, CSOs = Civil Society Organizations, DMA/DDA = Department of Municipal/ Development Affairs, DRD = Department of Rural Development.

The government stakeholder groups of TMAC/Town Elders, Department of Municipal/Development Affairs, local authorities, and political parties were classified as *definitive* types possessing all three attributes of *power*, *legitimacy*, *and urgency* for public water supply. A non-government stakeholder group of water users was classified as *demanding* with only an *urgency* attribute. In contrast, private sectors and the Department of Rural Development were classified as *discretionary* types with only *legitimacy*. Civil Society Organizations (CSOs were classified as *dependent types* possessing two attributes of *legitimacy and urgency*). Although water users are supposed to be the core stakeholder, they are identified as *demanding types*, *possessing* only *urgency* without *power* and *legitimacy*. So far, no regulatory frameworks protect their interests and secure their access to fresh water in the study area. Similarly, the private sector business groups are *legitimate stakeholders* without *power* and *urgency*. Local authorities and political parties act as the *definitive* type of stakeholders as they have high power, high urgency, and high legitimacy in town water supply services.

During the workshop, although the participants did not identify the *dormant, dominant, and dangerous types*, in the local context reality, the workshop participants agreed to identify the other stakeholder groups of religious organizations as *dormant types* with only *power* attributes, military institutions as *dominant type* with two attributes of *power and legitimacy*, and Ethnic Armed Organizations as *dangerous type* with both *power and urgency*. The analysis showed a high salience level of government stakeholders and low salience level of non-government stakeholders in this study area (Table 2 and Figure 4).

*3.3. Stakeholders' Perception of Water Supply Situation Causes and Consequences of Policy Changes*
3.3.1. Stakeholders' Perception of Water Supply Resources in Rakhine State

Key actors' statements/perceptions related to the current situation of water supply resources are presented in Table 3. Low water quantity and quality due to infrastructure problems and weak water resources management were the key issues expressed by the



stakeholders. There is no water quality control before supplying to the households, and a lack of access to water for many segments of the population. The workshop participants proposed actions for water conservation, as well as finding new and/or alternative water resources for the supply of the townships. They also appealed for investment in the water supply network, facilities, and infrastructure for water monitoring, treatment, and distribution. The workshop participants also suggested (1) increasing budget allocation by local state government, (2) installation of water meters, and (3) urgent development of town water supply master plan and the supply governance must be included, multi-stakeholders' involvement, formation of town water supply committee, formation of water users' committees, and increased supervision by responsible departments.

**Table 3.** Stakeholders' views on current state of freshwater management in Rakhine.

| Type of Information | | Stakeholders' Views |
|---|---|---|
| Water current issues | Infrastructure problems: | <ul><li>Low water quantity distributed irregularly to water user households.</li><li>Unequal water distribution to water users due to technical errors in leveling for water distribution pipelines.</li><li>Shortage of electricity for pumping and low technology for water distribution pipelines system.</li></ul> |
| | Water resources: | <ul><li>The distributed water is getting polluted due to the weak water purification system and lack of water quality test.</li><li>Experienced in water shortage during summer by water users.</li></ul> |
| | Management: | <ul><li>Lack of regular maintenance for the whole water supply system due to the insufficient financial support by State Government.</li><li>The collected water tariff still very low due to the lack of water meter system.".</li><li>Lack of supervision in water distribution.</li></ul> |
| Suggestions for addressing water current issues | Infrastructure problems: | <ul><li>The standard infrastructures should be equipped particularly by installing one overhead tank for each Town Ward.</li><li>Water quality control system should be upgraded by construction of more water purification tanks and regular cleaning of water collection tanks and distribution pipelines.</li></ul> |
| | Water resources: | <ul><li>To improve the amount of distributed water, the current water sources should be conserved well and need to identify additional water sources towards increasing water catchment areas.</li><li>To do regular silt suction (dredging) from current water sources which are being polluted by animals and invaders and illegal farmers near water sources.</li></ul> |
| | Management: | <ul><li>State government should allocate sufficient budget for town water supply system.</li><li>Water meter system should be practiced for collection of full water tax payment from water users.</li><li>Urgent need to develop Town Water Supply Master Plan covering coordination, stakeholders' engagement, upgrading existing system, capacity building for water users and DMA/DDA staff.</li></ul> |
| Proposal for improved freshwater supply governance | | <ul><li>Coordinative multi-stakeholders' involvement, formation of town water supply committee.</li><li>formation of water users committees in each town ward, increased supervision by responsible departments.</li><li>Increased transparency and accountability of local government,</li><li>Increased public participation, and public awareness raising sessions for systematic water utilization.</li></ul> |

### 3.3.2. Stakeholders' Perception of Causes and Consequences of Institutional Policy Changes during the Armed Conflict

Table 4 shows information obtained from the interviews with multi-stakeholders explaining that internally displaced persons (IDPs) during armed conflict along with the

COVID-19 pandemic in 2020 has made the change in policy to prioritize the water supply to IDPs, which created impacts on irregular and insufficient water supply to existing water users. In addition to that, due to the COVID-19 pandemic, the policy priority has also been emphasized to supply freshwater to COVID-19 quarantine centers. This is because the water supply systems are not efficient to cope with emergencies or crises.

**Table 4.** Stakeholder perceptions on causes and consequences of institutional policy changes during armed conflict.

| Institutional Policies during Armed Conflict | Causes of Institutional Policy Changes | | Consequences of Institutional Policy Changes | |
|---|---|---|---|---|
| **Water Supply** | (1) | Increasing IDPs during Armed Conflict | (1) | Facing irregular and insufficient water supply due to institutional policy had been prioritized to provide water supply to IDPs and COVID-19 quarantine centers |
| | (2) | COVID-19 pandemic | | |
| **Water pricing and tax collection** | (1) | Lack of law enforcement taking action against irregular water taxpayers | (1) | Unable to apply the rules and regulations on water tax collection |
| | (2) | Irregular water supply with shortages during armed conflict | (2) | Increasing water pricing |
| | | | (3) | Unable to collect water tax from water users |
| **System maintenance** | (1) | Armed conflict and COVID-19 pandemic | (1) | Delay institutional policy, plan, and projects |
| | | | (2) | Lower quality of service than normal situation |
| | | | (3) | Delay required maintenance of freshwater supply system |

It was also found that during the armed conflict, due to the lack of law enforcement taking action against irregular water tax payees, the policy was changed to increase water prices to recover the deficit in the budget. Furthermore, all water development plans and policies for maintenance and management have been also delayed because of administrative structure changes or a lack of budget from the central government. As a result, the availability and quality of water supply services became lower than the normal situation in this study area.

*3.4. Town Water Supply Governance in the Armed Conflicting Area from the Multi-Stakeholders' Perspective*

Multi-stakeholders' perceptions of town water supply governance during armed conflict were assessed in terms of actors' perception of inter–intra interaction among water agencies, institutions, and community, water users' satisfaction level of the provided public water supply system, quality of water supply services, the current status of the public water supply management system, and access to information on the water supply system.

As presented in Table 5 below, the governmental actors were positive about the inter–intra interaction among water agencies, institutions, and communities, while all the non-government actors, including political parties, stayed neutral. Local government and departmental authorities were satisfied with their facilitation and support services to water users' access to the water supply. Still, other informal institutions also stayed neutral in this aspect. Similarly, most actors from governmental agencies are positive about the current state of water quality and supply capacity; other actors, such as water users and private sectors, stayed neutral again.

TMAC/Town Elders, DMA/DDA, DRD, and local authorities determined that they have easily accessed information on the water supply system. However, many actors such as water users and private sectors, CSOs, and political parties confided that there was limited access to water supply-related information. The results showed the distorted perception of water quality and supply availability in the state because of different interests and political status, responsibility, and powers. Informal institutions, water users, and private sectors seemed not to dare express their views and remained neutral, although they

were not satisfied with the current state of water supply governance, water quality, and distribution (Table 5).

**Table 5.** Stakeholders' Perception of Current State of Water Supply Governance in Rakhine State.

| Indicators | Assessed Variables | TMAC/Town Elders, DDA/DMA, DRD, and Local Authorities | Water Users and Private Business Sector | CSOs | Political Parties |
|---|---|---|---|---|---|
| Inter–Intra Interaction | Actors' Perception on Inter–Intra Interaction | Positive | Neutral | Neutral | Neutral |
| Access to town water supply services | Satisfaction Level of Access to Town Water Supply Services | Positive | Positive | Positive | Positive |
| Quality of water supply service | Opinion on the Quality of Water Supply Service | Positive | Positive | Positive | Positive |
| Satisfaction | Level of Satisfaction on Freshwater Supply Management System | Positive | Neutral | Positive | Positive |
| Access to information | Level of Access to Information on Freshwater Supply System | Positive | Neutral | Neutral | Neutral |

## 4. Discussion

### 4.1. Lack of Socio-Technical and Financial Capacity in Public Water Supply Governance

The research findings showed that the perspectives of multi-stakeholders raised many concerns on water supply systems in Rakhine, including water quality, risk of water supply shortage during political emergencies, and natural disasters. During the armed conflict period, as the Rakhine State Government lacked the financial capacity to invest in basic infrastructures and socio-technical support for the public water supply system, there was no effective and efficient water supply quantity and quality control to address the prevailing issues and negative consequences of institutional policy changes on water supply, water tax collection, and system maintenance for sustainability.

It is consistent with the previous research finding in 2015, as many public water supply systems showed their poor capacities due to under maintenance and lack of funds for operation [32], and in line with the previous assessment finding in 2018, as a combination of poverty, water scarcity, armed conflict, and warfare has produced serious challenges for both water supply and sanitation [33].

The findings from this study contribute to making richer the knowledge and understanding of water supply governance by confirming that "there are weak water management systems in conflicting areas [9] and "armed conflicts directly or indirectly affect water management systems [10]". The implications of these findings further support "poor governance and bad water management with increasing internally displaced persons, political and civil conflicts and lack of coordination and interaction among water institutions exacerbate the problems of water [34]".

### 4.2. State-Led Water Management Versus the Role of Multi-Stakeholders in Public Water Supply Governance

Many dynamic powers affect the process of equal and transparent water supply governance. There was the participation of different stakeholders in our participatory workshops; however, the non-formal and non-state actors remained neutral in every aspect of the water supply governance assessment. They spoke more during the interviews, which they expressed about current issues of water supply governance in their state. The assessment of town water supply governance performance through the participatory workshops and interviews provided us with different perspectives about the views of different stakeholders and main water actors on the governance performance. The governmental stakeholders tended to be positive about all components of governance performance. At the same time, CSO and private water users stayed neutral when rating inter–intra interaction among water institutions and stakeholders, access to information, and satisfaction with the current town water supply service. The neutrality in this political and armed conflict is understandable.

The findings showed a high salience level of government stakeholders and a low salience level of non-government stakeholders, which highlights the reality of state-led water management in the governance structure of public water supply. It also depicted a picture of the power dynamic among different types of actors and institutions for public water supply governance in armed conflicting areas of Rakhine State. Although water supply governance is controlled by the state/governmental agencies, there are many other "powerful" actors influencing water supply regulations and distribution, such as *dormant actors* such as religious organizations, *dangerous actors* such as military institutions, and *dominant actors* such as Ethnic Armed Organizations.

All armed conflicting countries around the world have institutional structures formed in both central government ministries level and local/provincial level institutions with the top-down management system [11]. However, this structure does not function because there are too many other interests and power dynamics of many informal organizations and institutions mentioned above that can mobilize and influence the community resulting in water supply regulations and distribution. This governance structure often neglects the coordinated and participatory approach, which takes into account the participation, voices, needs, and interests of the private sector, community-based institutions, and water users [12].

To ensure equal access to water among water users, especially the voiceless or powerless groups, the emergence of community self-regulated governance becomes critical in this conflicting area of Rakhine State. This appeals to the interventions of international development and peacebuilding organizations to promote the establishment of self-regulated governance in this difficult context of a complex policy design in order to facilitate user autonomy [5] and to enhance adaptive governance for the community learning and adaptation to social-political change over time [6].

Governance of self-management and self-regulation should be at the heart of a solution to water resources management in this area [35], and full community participation is mainly required for changing the sector's approach from supply-driven to demand-responsive [36]".

## 5. Conclusions

This research depicts a rich picture of water supply governance problems in the armed conflicting area in Rakhine State, Myanmar. Although the water supply system is managed and governed by the state water authorities with the involvement of many administrative, political, and sectoral technical agencies and organizations, the non-formal community organizations such as ethnic armed military and religious institutions also have a strong interest in water supply and are considered dangerous actors in the water supply governance process.

The findings showed that the diverse water actors held different perspectives and perceptions of water supply quality and quantity because of their different power holdings and political and economic interests. The state actors seemed biased on their positive performance, demonstrating their satisfaction with the current water supply governance in Rakhine State. The community, private sectors, and household water users instead showed their dissatisfaction with the quality and quantity of the current water supply system. Still, they stayed neutral about the water supply governance performance. The results show the complexity and dynamics of water actors' powers and interests in the armed conflicting areas. In addition, there is a lack of socio-technical and financial capacity for the investment and maintenance of water distribution and collection infrastructure and facility, as well as water quality and quantity monitoring and evaluation. Thus, community self-regulated water governance becomes critical for development and peacebuilding organizations working in conflicting areas to promote adaptive governance for the community learning and adaptation to social-political and environmental change over time.

**Author Contributions:** T.K.: Conceptualization. Research design, investigation, data curation; formal analysis, visualization, writing—original draft; writing—review and editing. T.P.L.N.: Conceptualiza-

tion, research design, validation, writing—original draft, writing—review and editing, supervision. All authors have read and agreed to the published version of the manuscript.

**Funding:** This research received no external funding.

**Informed Consent Statement:** Informed consent was obtained from all participants involved in the study.

**Data Availability Statement:** Data is contained within the article.

**Acknowledgments:** The authors acknowledge the data support from the State Department of Municipal/Development Affairs and the respective Township Departments in the study area of Rakhine State, Myanmar.

**Conflicts of Interest:** The authors declare no conflict of interest.

## Appendix A. Research Indicators, Variables, and Measurement Scale Points

**Table A1.** Indicators and variables used in the structured interviews and surveyed questionnaires for meas-uring stakeholders' perceptions of water supply governance.

| Indicators | Variables | Scale of Measurement |
|---|---|---|
| 1. Inter–Intra Interaction | 1.1 Set up the freshwater supply-related policies and priorities<br>1.2 Well informed to all actors about freshwater-related acts, regulations and laws<br>1.3 Regular relationship between actors and staff from the water supply system<br>1.4 Regular relationship between actors and water supply government agencies<br>1.5 Regular relationship between actors and local communities | 1 (Definitely Not)<br>5 (Definitely) |
| 2. Access to Water Supply Services | 2.1 Water supply department facilitated in freshwater accessibility<br>2.2 Local government helped to access freshwater resource<br>2.3 Informal institutions facilitated in freshwater use | 1 (Definitely Not)<br>5 (Definitely) |
| 3. Quality of Water Supply Service | 3.1 Staff and different stakeholders have regular monthly interactions<br>3.2 Permanent staff are working for town water supply related departments<br>3.3 Responsible staff are working all the weekdays<br>3.4 Staff are well equipped with suitable transportation facilities<br>3.5 Town water supply committee and other actors meet frequently | 1 (Strongly Disagree)<br>5 (Strongly Agree) |
| 4. Satisfaction | 4.1 Satisfaction on current status and functions of the freshwater supply management system | 1 (Very Dissatisfied)<br>5 (Very Satisfied) |
| 5. Access to Information | 5.1 Difficulty or easiness to access information on the freshwater supply management system | 1 (Very Hard to Access)<br>5 (Very Easy to Access) |

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
