# Peer review of "An Assessment of Water Supply Governance in Armed Conflict Areas of Rakhine State, Myanmar"

_water, doi:10.3390/w14182930_

Round 1

Reviewer 1 Report

This paper discusses the management of stakeholders in Myanmar’s water supply systems using the salience model and institutional analysis. This paper examines a topic that is both timely and intriguing: the provision of water in conflict zones. It focuses on stakeholders’ diverse interests and expectations in socially and politically sensitive areas. In addition to this strength, this paper is well organized and written, making it simple to comprehend. However, this reviewer believes that this paper can be improved in the following areas:

a.       While this paper focused on water supply governance, it would be helpful for readers if the authors could elaborate on what “governance” refers to in the context of this paper.

b.       The authors are recommended to elucidate this paper’s stated problem and the contributions to the existing body of literature from a global perspective to demonstrate this paper’s importance. Noting that water is one of the most essential and indispensable natural resources in particular armed conflict areas is not remarkably sufficient as it has been the conventional wisdom.

c.       The theoretical background is deemed insufficient. This reviewer is unsure whether this is due to the allowable limited space of this paper; if not, the authors may be advised to improve the literature in related areas.

d.       This reviewer suggests merging sections 3.2 (Public water supply system in the study area) and 2.1 (Study area), as section 3.2 describes existing conditions in the study area, which are not the study’s findings.

e.       The authors identified seven stakeholders. Could the authors explain how to map these identified stakeholders onto the salience diagram? Was there a consensus among respondents during the workshops?  

f.        The most critical way the authors can do to improve their paper is to elaborate more on the mapping results; for instance, why water users, who, according to this reviewer, should be the core stakeholder, are considered to have only urgency but not power and legitimacy, making them a demanding stakeholder. Are there no regulatory frameworks that protect their interests and secure their access to freshwater? On another front, religious organizations only have power but not legitimacy and urgency. Similarly, the private sector (who are they?) is a legitimate stakeholder without power and urgency. It is intriguing to learn that the local authorities and political parties act as the definitive stakeholders.  

g.       This reviewer argues that there is somehow a disconnect between the findings. The authors may find a way to better integrate the performance analysis results with the rest of this paper’s findings. For instance, the authors have defined five indicators to measure governance performance (note: How were these indicators defined) and found disagreements regarding the performance between the state actors and community, including the private sectors. What potential solutions might the authors propose?

Author Response

a.  This paper discusses the management of stakeholders in Myanmar’s water supply systems using the salience model and institutional analysis. This paper examines a topic that is both timely and intriguing: the provision of water in conflict zones. It focuses on stakeholders’ diverse interests and expectations in socially and politically sensitive areas. In addition to this strength, this paper is well organized and written, making it simple to comprehend.However, this reviewer believes that this paper can be improved in the following areas:

We would like to thank you for your valuable comments to help us improve the research paper. We have revised our paper highlighted in red in the manuscript according to your recommendation.

b. While this paper focused on water supply governance, it would be helpful for readers if the authors could elaborate on what “governance” refers to in the context of this paper”.

We have added under “Introduction” highlighted in red (Line 33-40).

c. “The authors are recommended to elucidate this paper’s stated problem and the contributions to the existing body of literature from a global perspective to demonstrate this paper’s importance. Noting that water is one of the most essential and indispensable natural resources in particular armed conflict areas is not remarkably sufficient as it has been the conventional wisdom.”

The research’s stated problem and how it has a potential contribution to the existing literature has been improved in the Introduction section (Line 126-137)

d. “The theoretical background is deemed insufficient. This reviewer is unsure whether this is due to the allowable limited space of this paper; if not, the authors may be advised to improve the literature in related areas”.

The theoretical background has been improved as suggested

e. “This reviewer suggests merging sections 3.2 (Public water supply system in the study area) and 2.1 (Study area), as section 3.2 describes existing conditions in the study area, which are not the study’s findings.”

Many thanks, and we have already merged section 3.2 to section 2.1 as suggested.

f. “The authors identified seven stakeholders. Could the authors explain how to map these identified stakeholders onto the salience diagram? Was there a consensus among respondents during the workshops?”

More information has been added to explain better how identified stakeholders was mapped onto the salience diagram under 3.2.1 Public water supply stakeholders and their salience.”

g. “The most critical way the authors can do to improve their paper is to elaborate more on the mapping results; for instance, why water users, who, according to this reviewer, should be the core stakeholder, are considered to have only urgency but not power and legitimacy, making them a demanding stakeholder. Are there no regulatory frameworks that protect their interests and secure their access to freshwater? On another front, religious organizations only have power but not legitimacy and urgency. Similarly, the private sector (who are they?) is a legitimate stakeholder without power and urgency. It is intriguing to learn that the local authorities and political parties act as the definitive stakeholders.”

Thank you, more explanation about the mapping results of different types of stakeholders’ salience has been added under “3.2.1 Public water supply stakeholders and their salience”, as suggested.

 h. “This reviewer argues that there is somehow a disconnect between the findings. The authors may find a way to better integrate the performance analysis results with the rest of this paper’s findings. For instance, the authors have defined five indicators to measure governance performance (note: How were these indicators defined) and found disagreements regarding the performance between the state actors and the community, including the private sectors. What potential solutions might the authors propose? “

The water supply governance assessment in this paper was made through integrated methods and several research stages, from participatory workshops to interviews. We assess the water supply governance assessment by examining different aspects through stakeholders’ lens: their perceptions of the current state of freshwater management, their salience, their perspectives of policy changes, and finally, their rating governance performance of inter/intra interaction, access to information, the satisfaction of stakeholders, etc. The five indicators defined were only used for rating the governance performance. We have restructured and arranged the data analysis section and connected all findings in the results and discussions.

Reviewer 2 Report

The authors present the results of an investigation into water supply governance in a former armed conflict zone (Myanmar). The area discussed seems to have been a former conflict zone as the authors interviewed government officials and others who potentially would be excluded from an active conflict zone. The issues raised by the government officials, public, local organizations and corporations did not directly address the issue of operating a water supply in wartime--government seemed to be active and in place, infrastructure seemed to be present and (more or less) operational, and civil society was engaged. Nevertheless, the authors identify a number of issues that are applicable not only to conflict zones but to developing countries more generally. 

That said, beginning on lines 10, 24 and 27, the term "conflicting zones" should be "conflict zones".

References are needed on lines 57 (Myanmar National Water Policy), 69 (UNDP Local Governance Mapping), 73 (International Crisis Group), and 123-125 (basin and socio-economic data).

Table 2 suggests that the respondents were probably atypical of the population of Rakhine State being older males with a university education. Perhaps the authors could comment on the representative nature of the respondents?

In section 3.2, lines 212-221, the analysis of secondary data is mentioned but does not seem to have been presented in the results section?

Line 275, please define CSOs (civil society organizations) at the first mention of the acronym (it is defined on line 311).

Table 4, under the second heading, "Dragging" should be "Dredging".

Table 5 and Line 350, "payee" probably should be "payer"?

The paper should be reviewed for standard English usage. There are several areas where quotations are suggested, but no opening quotes seem to be present.

Author Response

The authors present the results of an investigation into water supply governance in a former armed conflict zone (Myanmar).The area discussed seems to have been a former conflict zone as the authors interviewed government officials and others who potentially would be excluded from an active conflict zone. The issues raised by the government officials, public, local organizations and corporations did not directly address the issue of operating a water supply in wartime--government seemed to be active and in place, infrastructure seemed to be present and (more or less) operational, and civil society was engaged.

Nevertheless, the authors identify a number of issues that are applicable not only to conflict zones but to developing countries more generally.

We would like to thank you for your valuable comments to help us improve the research paper. We have revised our paper according to your recommendation as follows.

a. “ beginning on lines 10, 24 and 27, the term "conflicting zones" should be "conflict zones"

Change to “conflict zone” as suggested.

b. “References are needed on lines 57 (Myanmar National Water Policy), 69 (UNDP Local Governance Mapping), 73 (International Crisis Group), and 123-125 (basin and socio-economic data).”

References have been cited and added

 c. “Table 2 suggests that the respondents were probably atypical of the population of Rakhine State being older males with a university education. Perhaps the authors could comment on the representative nature of the respondents?”

It is typical in Myanmar; male and senior people dominate the administrative and political positions, thus similarly in Rakhine. Comments has been added for these findings.

 d. “ In section 3.2, lines 212-221, the analysis of secondary data is mentioned but does not seem to have been presented in the results section? So need to remove about secondary data “

Thank you, the research was done mainly based on the primary data. Some secondary data was used to describe the town's water supply system and water management structure. We have removed the secondary data in the data collection section.

e. ‘Line 275, please define CSOs (civil society organizations) at the first mention of the acronym (it is defined on line 311)”.

“Civil society organizations” has now been fully mentioned at first introduction.

 f. “Table 4, under the second heading, "Dragging" should be "Dredging"

Changed to Dredging

g. “ Table 5 and Line 350, "payee" probably should be "payer"?

Revised

h. “ The paper should be reviewed for standard English usage. There are several areas where quotations are suggested, but no opening quotes seem to be present.”

Thank you. The paper has been checked in English carefully. There are no quotes presented in the papers.

Round 2

Reviewer 1 Report

This reviewer congratulates the authors on their excellent work. Their revised paper addresses most of the reviewer's comments on the earlier draft. Based on this reason, this reviewer suggests acceptance with minor revisions. The authors are encouraged to recheck their work for any grammatical and typographical errors (e.g., [3131, line 277], Table 6 [showed]). In addition, this reviewer suggests that the paragraph that references Table 2 be placed after the heading and updated accordingly; for example, "As presented in Table 2, the Chi-square...."

Author Response

We would like to thank the Reviewer for the constructive comments and appreciation of our careful revision.

As requested, we have carefully checked the grammatical and typographical errors of the whole document.  In addition, the paragraph that references Table 2 is now placed after the heading and updated.